# Awake Prone Positioning, High-Flow Nasal Oxygen and Non-Invasive Ventilation as Non-Invasive Respiratory Strategies in COVID-19 Acute Respiratory Failure: A Systematic Review and Meta-Analysis

**DOI:** 10.3390/jcm11020391

**Published:** 2022-01-13

**Authors:** Benedikt Schmid, Mirko Griesel, Anna-Lena Fischer, Carolina S. Romero, Maria-Inti Metzendorf, Stephanie Weibel, Falk Fichtner

**Affiliations:** 1Department of Anesthesiology, Intensive Care, Emergency and Pain Medicine, University Hospital Wuerzburg, 97080 Wuerzburg, Germany; schmid_b@ukw.de (B.S.); weibel_s@ukw.de (S.W.); 2Department of Anaesthesiology and Intensive Care, University of Leipzig Medical Services, 04103 Leipzig, Germany; Mirko.Griesel@medizin.uni-leipzig.de (M.G.); Anna-Lena.Fischer@medizin.uni-leipzig.de (A.-L.F.); 3Department of Anaesthesiology, Intensive Care, Emergency and Pain Medicine, General University Hospital, 46014 Valencia, Spain; carolinasoledad.md@gmail.com; 4Cochrane Metabolic and Endocrine Disorders Group, Institute of General Practice, Medical Faculty of the Heinrich-Heine-University Düsseldorf, 40225 Duesseldorf, Germany; Maria-Inti.Metzendorf@med.uni-duesseldorf.de

**Keywords:** respiratory failure, non-invasive ventilation, high-flow nasal cannula, awake prone positioning, COVID-19, systematic review

## Abstract

Background: Acute respiratory failure is the most important organ dysfunction of COVID-19 patients. While non-invasive ventilation (NIV) and high-flow nasal cannula (HFNC) oxygen are frequently used, efficacy and safety remain uncertain. Benefits and harms of awake prone positioning (APP) in COVID-19 patients are unknown. Methods: We searched for randomized controlled trials (RCTs) comparing HFNC vs. NIV and APP vs. standard care. We meta-analyzed data for mortality, intubation rate, and safety. Results: Five RCTs (2182 patients) were identified. While it remains uncertain whether HFNC compared to NIV alters mortality (RR: 0.92, 95% CI 0.65–1.33), HFNC may increase rate of intubation or death (composite endpoint; RR 1.22, 1.03–1.45). We do not know if HFNC alters risk for harm. APP compared to standard care probably decreases intubation rate (RR 0.83, 0.71–0.96) but may have little or no effect on mortality (RR: 1.08, 0.51–2.31). Conclusions: Certainty of evidence is moderate to very low. There is no compelling evidence for either HFNC or NIV, but both carry substantial risk for harm. The use of APP probably has benefits although mortality appears unaffected.

## 1. Introduction

As of November 2021, there have been more than 250 million confirmed SARS-CoV-2 cases with more than five million deaths globally [1]. Although many infections present with no or mild symptoms, more severe cases can be characterized by hypoxemia, which then often leads to hospitalization. Reports on large registry datasets found hospitalization rates of 14.0% and the need for ICU admission in 2.3–5% of all confirmed cases [2,3]. Among the most dangerous manifestations of the SARS-CoV-2 infection is the progression of respiratory insufficiency to acute respiratory distress syndrome (ARDS). Out of all hospitalized patients, 17% were found to require invasive mechanical ventilation in an analysis of health insurance data [4].

To alleviate the burden of disease for both patients and healthcare systems during this pandemic, researchers have been eager to find the best strategies for the therapeutic management of COVID-19-associated respiratory failure. Early on, COVID-ARDS was suspected to be pathophysiologically and morphologically different from known (infectious or non-infectious) ARDS [5,6]. However, later and larger studies did not confirm these early theories [7,8,9]. As a result, the principles of respiratory therapy for COVID-19 associated respiratory failure were similar to those of “conventional” ARDS [10].

After the initial trend towards early intubation to reduce aerosols, in the context of an ongoing global health crisis and in an alleged attempt to spare resources by both questioning work-intensive procedures and preventing disease progression as early as possible, research also started focusing on non-invasive ventilation (NIV) and high-flow nasal cannula oxygen administration. NIV can be applied by a face mask or helmet, both of which have to be fixed tightly to prevent pressure loss. A continuous positive airway pressure is applied and shall prevent collapse of the alveoli and atelectasis. HFNC promises to increase patient comfort, require less attention by healthcare professionals than conventional non-invasive ventilation, and may help avoid invasive ventilation, too [11,12]. Non-invasive ventilation strategies have been found to reduce mortality and lower the risk of intubation [13]. So far, no evidence-based recommendations favor either HFNC or conventional non-invasive ventilation (NIV) [14]. Awake prone positioning (APP) was hypothesized to lead to similar physiological improvements (e.g., reduction of ventilation-perfusion mismatches) as prone positioning in sedated, intubated patients [15,16], but perhaps do so in an earlier stage of the disease, thus slowing down or avoiding progression to more severe conditions. Case series and non-randomized controlled trials (non-RCTs) suggested feasibility of APP and a potential clinical benefit for patients with hypoxemic respiratory failure [17,18,19]. For both research questions (i.e., the value of HFNC or NIV and APP in COVID-19-associated respiratory failure), we prospectively registered protocols for systematic reviews on the matter. Here, we report and analyze the available evidence on HFNC vs. NIV and APP vs. standard of care (SoC) on COVID-19-associated respiratory failure. We provide meta-analyses for pre-defined, patient-centered core outcomes and assess certainty of evidence and possible resulting recommendations using systematic, validated approaches.

## 2. Methods

To assess the efficacy of awake prone positioning (APP) and the use of high-flow nasal oxygen or non-invasive ventilation (NIV), respectively, on COVID-19-induced respiratory failure, we included randomized controlled trials. Non-RCT designs were not eligible in case that RCT(s) on the matter were available. Thus, we excluded controlled, non-randomized studies and observational studies. We included full-text journal publications, preprint articles, abstracts publications and results published in trial registries if they provided information in sufficient detail. The methodology was predefined and registered in two separate PROSPERO protocols with any deviations pointed out below (APP/SoC: CRD42021261862 [20]; HFNC/NIV: CRD42021230825 [21])

We conducted two separate systematic searches for each topic in the following sources from initiation of those databases to the date of search without restrictions on the language of publication:Cochrane COVID-19 Study Register (COVID-19.cochrane.org), comprising Cochrane Central Register of Controlled Trials, MEDLINE (PubMed), Embase, ClinicalTrials.gov, WHO International Clinical Trials Registry Platform, and medRxiv (last searched on 26 October 2021 for both topics);WHO COVID-19 Global literature on coronavirus disease (https://search.bvsalud.org/global-literature-on-novel-coronavirus-2019-ncov) (last searched on 26 October 2021 for both topics);Web of Science (Science Citation Index Expanded and Emerging Sources Citation Index) (last searched on 1 March 2021 for NIV and 7 June 2021 for APP);CINAHL (Cumulative Index to Nursing and Allied Health Literature) (last searched on 1 March 2021 for NIV and 7 June 2021 for APP).

For detailed search strategies, see Appendix A. We identified other potentially eligible studies or ancillary publications by searching the reference lists of included studies, relevant systematic reviews and meta-analyses. 

We included studies investigating adult patients with severe respiratory failure due to COVID-19 infection according to the WHO clinical progression scale [22]. This includes patients with the need for high-flow nasal oxygen, non-invasive ventilation, or invasive ventilation. COVID-19 infection had to be confirmed by reverse-transcription polymerase chain reaction (RT-PCR) or highly suspected by clinical presentation of the patient.

With regard to awake proning, full prone or 135° prone positioning were acceptable interventions. We did not define any restrictions on the duration of proning per day. Standard of care as the comparator was defined as 90° or supine positioning and otherwise all diagnostic and therapeutic interventions considered necessary at the treating physician’s discretion.

With regard to non-invasive ventilation, we deviated from the initially registered protocol: We focused on the highly relevant question from daily clinical practice about the efficacy and safety of HFNC (intervention) compared to NIV (standard care) as a non-invasive strategy of respiratory support. We did not include trials comparing different ventilator settings within HFNC or NIV, nor comparing simple oxygen insufflation or invasive ventilation to either HFNC or NIV. Additionally, we excluded comparisons of NIV vs. awake proning because there is no rationale not to combine both. The outcome set and the analysis specifications were adapted to the latest version of the APP protocol [20].

We evaluated core outcomes in accordance with the Core Outcome Measures in Effectiveness Trials (COMET) initiative for COVID-19 patients [22]. These outcomes were in part modified to better represent the nature of the non-pharmacological interventions investigated in this meta-analysis. Additional outcomes specific to our research question were also added. The following outcome set served as the basis for our analyses:

Main outcomes:-All-cause mortality at day 28, day 60, and time-to-event, and at hospital discharge-Clinical status at day 28, day 60, and up to the longest follow-up, including:
-worsening of clinical status: patients with clinical deterioration or death-improvement of clinical status: patients discharged alive without clinical deterioration or death-quality of life, including fatigue and neurological status, assessed with standardized scales at up to 7 days, up to 28 days and longest follow-up available
-Serious adverse events during the study period, defined as number of patients with any event-Adverse events (any grade) during the study period, defined as the number of patients with any event

Additional outcomes:-Clinical status at day 28, day 60 and up to longest follow up, including:
-worsening of clinical status: new need for invasive mechanical ventilation-improvement of clinical status:
-weaning or liberation from invasive mechanical ventilation-liberation from supplemental oxygen in surviving patients-ventilator-free days-duration to liberation from invasive mechanical ventilation-duration to liberation from supplemental oxygen

-Admission to intensive care unit at day 28-Duration of hospitalization-Skin lesions from proning measures

Two review authors independently identified studies possibly eligible for this review. If encountering disagreement, we consulted a third review author and resolved all discrepancies by discussion. Further, two review authors independently extracted data using a standardized data extraction form.

Two review authors independently assessed the risk of bias for RCTs using the Cochrane “Risk of Bias” tool 2 (RoB2) [23]. The effect of interest is the effect of assignment at baseline, regardless of whether the interventions were received as intended (intention-to-treat). Risk of bias was assessed for all results which were identified as one of the predefined outcomes of this review. The overall risk of bias for each outcome was summarized according to RoB2 guidance.

Random effects meta-analyses were performed with RevMan 5, as far as the patient characteristics (disease severity) and interventions (standard care and treatment settings) were comparable across studies. Dichotomous outcomes were recorded as number of events and total patient number in all study groups. Pooled risk ratios (RR) and 95% confidence intervals (95% CI) were calculated. Continuous outcomes were recorded as mean, standard deviation and total number of patients in each group. Comparisons of continuous outcomes were reported as the difference of means (MD) and the corresponding 95% CI. Statistical heterogeneity was defined as *p* < 0.1 for the Chi^2^ test of heterogeneity or I^2^ ≥ 50%. We had planned to explore heterogeneity by subgroup analysis to calculate RR or MD in conjunction with the corresponding CI for each subgroup if sufficient studies had been available (at least 10 studies per outcome). For the current review, there were not enough studies available.

Certainty of evidence was assessed using the Grading of Recommendations Assessment, Development and Evaluation (GRADE) approach. Summary of findings tables were created using the MAGICapp software (Norwegian MAGIC Evidence Ecosystem Foundation powered by UserVoice Inc.; MAGICapp. Brønnøysund, Norway) for the proning part and GRADEpro APP (GRADEpro GDT: GRADEpro Guideline Development Tool [Software]. McMaster University and Evidence Prime, 2021. Available from gradepro.org (accessed on 6 January 2022)) for the HFNC/NIV part. According to the five domains used in the GRADE approach, we downgraded the certainty of evidence of each individual outcome as follows:-serious (−1) or very serious (−2) risk of bias-serious (−1) or very serious (−2) inconsistency-serious (−1) or very serious (−2) uncertainty about directness-serious (−1) or very serious (−2) imprecision of the data-serious (−1) or very serious (−2) probability of reporting bias

The grade of evidence is then assigned one of the labels “high”, “moderate”, “low”, or “very low”.

## 3. Results

### 3.1. Study Selection

The literature search produced 4985 (HFNC vs. NIV: 3696/APP: 1289) records and after deduplication resulted in 3938 (HFNC vs. NIV: 2914/APP: 1024) records. After title and abstract screening, 22 records (APP: 13/HFNC vs. NIV: 9) were identified as potential RCTs. We retrieved 22 full-text articles including registry entries (HFNC vs. NIV: 9/APP: 13). Of those, 17 records were excluded (HFNC vs. NIV: 4, no results published; 2 duplicates/APP: 2, wrong population; 9, meta-trial), resulting in 5 records (APP: 2/HFNC vs. NIV: 3) for data extraction. The detailed search process is outlined in Figure 1.

### 3.2. Study Characteristics

#### 3.2.1. HFNC vs. NIV

We included three unblinded RCTs for the direct comparison of HFNC vs. NIV. Data from the largest study by Perkins et al., a pragmatic multicenter multinational adaptive three-armed RCT with 797 patients allocated to either HFNC or NIV, is currently available from a preprint version only [25]. We omitted patients from the “O_2_-insufflation only” arm of this preprint. Apart from that, there were two smaller conventional two-armed RCTs: the Italian multicenter study by Grieco et al. [26] and the Indian single-center study by Nair et al. [27], both allocating 109 patients and available as peer-reviewed publications.

All three trials enrolled hospitalized patients with similar disease severity at the verge of WHO stages 5 and 6, i.e., with severe respiratory failure (means of PaO_2_/FiO_2_ ratio ranging from 102 to 139 and tachypnea present) but invasive ventilation not yet imminent. All included RCTs conducted standard care according to their local clinical practice: dexamethasone was given (Grieco, Perkins), as well as remdesivir (Grieco) and since January 2021 Tocilizumab (Perkins). Every study allowed or encouraged awake prone positioning for both treatment groups, although Perkins (HFNC group: 58.3% vs. NIV group: 54.5%), Grieco (HFNC group: 60% vs. NIV group: 0%), and Nair (no exact data on APP rate; HFNC group: “almost all[...]” vs. NIV group: “[...]noncompliant[...]”) reported substantial differences of APP rate between HFNC and NIV groups, which could have introduced bias. Overall, there were no relevant differences between the treatment groups in terms of adherence to the intervention protocol (relative number of patients that received the intended intervention: Perkins: 92.1% HFNC vs. 91.6% NIV; Grieco: 87% HFNC vs. 91% NIV; Nair: not reported).

The recruitment periods overlapped well, patient characteristics including disease severity with PaO_2_/FiO_2_ ratios were similar across studies (indicating moderate ARDS) and standard care and treatment settings were sufficiently homogenous.

Of our predefined main outcomes, all three studies reported in-hospital mortality. Other outcomes were presented on different scales or for different time points. Table 1 summarizes the characteristics and intervention details of the included studies along with the reported outcomes.

#### 3.2.2. APP

We included two RCTs on APP in this systematic review. The study by Ehrmann et al. [28] was designed and conducted as a prospective meta-analysis of six individual multicenter RCTs. As its results were reported and published in a combined publication, we did not assess the individual national RCTs in this systematic review. The second included multicenter RCT by Rosén et al. [29] contributed 75 of the overall 1196 patients included in this review. In both trials, participants were treated in an in-hospital setting. The intervention of APP intended to last for as long as possible in the trial by Ehrmann, whereas Rosén defined a proning duration of at least 16 h per day. This goal was not met (median prone time 9.0 h, IQR 4.4–10.6). Moreover, participants in the control group were also exposed to prone positioning for a median of 3.4 h per day (IQR 1.8–8.4). In the trial by Ehrmann, durations of proning varied widely between the participating countries. Mean proning duration in Spain was 1.7 h/day (SD 1.2), whereas in the Mexican sub-trial patients spent 9.0 (3.2) hours in a prone position. Proning in the control groups occurred only to a very small extent.

Of our predefined main outcomes, both studies reported mortality at day 28 as well as need for intubation until day 28. Table 1 summarizes the characteristics of the included studies and the reported outcomes.

### 3.3. Risk of Bias

#### 3.3.1. HFNC vs. NIV

We assessed risk of bias and methodological quality for both RCTs using the RoB2 tool. In the overall bias assessment, we rated no outcome at low risk, three outcomes with some concerns and 14 with high risk of bias. This was mostly due to the unblinded study design affecting deviations from intended intervention (Perkins, Nair, one outcome Grieco), but also problems in the randomization process (Nair), selection of the reported result (Grieco). Detailed assessment of risk of bias for each outcome is summarized in Table 2.

#### 3.3.2. APP

In total, the studies contributed 11 relevant outcomes. Due to the nature of the interventions, neither patients nor healthcare providers were able to be blinded in the included trials, which led at least to some concerns in every reported outcome. In the overall bias assessment, we rated one outcome at low risk, five outcomes with some concerns and five with high risk of bias. The outcomes we rated to be at high risk of bias were all from Rosén et al. due to the high amount of awake prone positioning in the control group. Detailed assessment of risk of bias for each outcome is summarized in Table 2.

### 3.4. Effects of the Intervention

#### 3.4.1. HFNC vs. NIV in COVID-19 Acute Respiratory Failure

The main outcome of in-hospital mortality (to longest follow-up) was reported by the three included RCTs (RR 0.92; 95% CI 0.65–1.33, 986 patients from 3 studies, very low certainty), while two of the studies further reported mortality up to day 30 (RR 1.14; 95% CI 0.86–1.51, 902 patients from 2 studies, very low certainty), hence we do not know whether HFNC compared to NIV increases or decreases the risk of death. We downgraded certainty of evidence for both mortality outcomes due to serious risk of bias, serious imprecision, and indirectness.

Perkins provided data for another main outcome, intubation or death at day 28. HFNC compared to NIV may increase intubation or death (RR 1.22, 95% CI 1.03–1.45, 791 patients from 1 study). Certainty was low due to serious risk of bias and serious indirectness. Perkins and Grieco reported intubation within 28 and 30 days. We do not know if HFNC compared to NIV increases the risk of intubation (RR 1.34, 95% CI 1.00–1.80, 900 participants from 2 studies, very low certainty). Downgrading to very low certainty of evidence was due to serious risk of bias, imprecision, and indirectness. Adverse events were reported by Perkins and Grieco, although a predefined selection of events relevant in respiratory and intensive care medicine was made by the latter. As we deemed this selection still useful, we extracted AEs from both trials in the only way the reporting allowed to facilitate comparison: total events per patient in each arm. We do not know if HFNC vs. NIV increases or decreases AEs (Perkins: NIV 200/380 vs. HFNC 157/417; Grieco: 37/54 vs. 70/55, very low certainty). The downgrading to very low certainty is based on serious risk of bias, inconsistency, and indirectness. Length of Hospital stay with a follow-up of 30 days was reported by Perkins and may be longer in the HFNC group (MD 1.9, 95% CI 0.75 less to 4.55 more; baseline 16.4 in NIV group, low certainty). We downgraded due to serious risk of bias, imprecision, and indirectness. Respirator-free days defined as free from invasive ventilation, HFNC, and NIV in one study may have been decreased by HFNC (MD 2.0 days, 95% CI 6.16 less to 2.13 more, with baseline 15 days in the NIV group, low certainty). We downgraded here for very serious imprecision.

#### 3.4.2. APP in COVID-19 Acute Respiratory Failure

The main outcome of mortality at 28 days was reported by both studies (*Rosén* reported mortality at 30 days). Prone positioning may have little to no effect on mortality at 28 days (RR 1.08, 95% CI 0.51–2.31, 1196 participants from two studies). The certainty of evidence is low for this outcome due to serious risk of bias and serious imprecision.

*Ehrmann* provided data for another main outcome, the combined risk of intubation or death until day 28. APP probably decreases the risk for this outcome (RR 0.86, 95% CI 0.75–0.98, 1121 participants from 1 study). Certainty of evidence was moderate due to serious risk of bias.

With moderate certainty of evidence due to serious risk of bias, prone position compared to standard of care probably decreases the rate of intubation (RR 0.83, 95% CI 0.71–0.96, 1196 participants from 2 studies) and probably increases the time of weaning of HFNC slightly (MD 0.9 days, 95% CI 0.35–1.45, 1121 participants from 1 study), but probably has little or no effect on length of hospital stay (difference of means −0.2 days, 95% CI −1.35–0.96, 1196 participants from 2 studies). With low certainty of evidence due to serious risk of bias and imprecision, APP may decrease the days free of HFNC slightly (difference of means 2 days, 95% CI 0.13–3.87, 50 participants from 1 study). With regard to the occurrence of skin lesions due to the intervention, we are uncertain whether APP increases or decreases the risk for this outcome (RR 0.5, 95% CI 0.16–1.56, 1196 participants from 2 studies).

Effects of the interventions are detailed in the summary of findings tables, Table 3 for HNFC vs. NIV and Table 4 for APP.

## 4. Discussion

This review aimed to assess the efficacy and safety of different non-invasive strategies of respiratory support for the treatment of COVID-19.

So far, substantial recommendations on the use of non-invasive respiratory techniques in COVID-19 acute respiratory failure could not be made due to a lack of evidence. Nevertheless, previous studies in non-COVID-19 acute respiratory failure have shown beneficial outcomes for patients treated with HFNC or NIV [30]. Similarly, prone positioning has been a key part of non-COVID-19 ARDS treatment strategies for years [31]. Considering the urgent need for effective and efficient treatment strategies in the face of an ongoing pandemic, HFNC/NIV and APP could benefit both patients and healthcare systems alike. Obviously, the dynamic of the pandemic with changes in standard of care, vaccination as well as virus variants leaves us with the problem of transferability to present and future patients.

Regarding the question whether HFNC is superior to NIV as non-invasive respiratory support in COVID-19 patients suffering acute respiratory failure, we identified three RCTs (Grieco 2021, Nair 2021, Perkins 2021). These had been conducted in three different countries, including two high-income (United Kingdom, Italy) and one lower-middle-income country (India) [32]. The largest study, Perkins, included a considerable share (about 15%) of patients without laboratory-confirmed COVID-19.

The available evidence on APP is based on two RCTs, of which Ehrmann et al. was designed and conducted as a prospective meta-analysis/multi-national meta-RCT [28] and contributed more than 90% of the patients included in this review. Both trials recruited patients during comparable time periods and relevant accompanying treatments as part of SoC were similar (e.g., administration of glucocorticoids in a large proportion of all patients). The included trials reported difficulties applying the intended intervention in a standardized manner: Ehrmann et al. aimed to apply awake prone positioning for as long as possible each day. However, the absolute durations of actual APP varied widely between the involved countries (1.7 ± 1.2 h in Spain vs. 9.0 ± 3.2 h in Mexico; overall trial: 5.6 ± 4.4 h). Longer APP durations were associated with greater treatment success. APP in the SoC control group was little to none, which was an issue the trial by *Rosén* was facing. There, the authors described increasing durations of APP in patients from the control group, which eventually also led to the termination of the trial.

The five RCTs on non-invasive respiratory strategies took place during the second half of 2020 and first half of 2021, hence presumably different variants of SARS-CoV-2 were prevalent and vaccination programs, not yet rolled out at full scale or at all. Although basic pathophysiology might not, pharmacologic and non-pharmacologic treatment recommendations and practice in COVID-19 have changed over time. We cannot quantify possible indirectness but consider it to be of minor influence.

The certainty of the evidence regarding our predefined outcomes was downgraded for all outcomes, mostly due to serious risk of bias, serious imprecision and/or serious indirectness.

In our analysis of the study data on benefits and harms of HFNC compared to NIV, there remains uncertainty whether HFNC slightly increases or decreases mortality (different mortality outcomes, i.e., in-hospital mortality, mortality up to day 30 with different effect directions). Nonetheless, NIV may reduce the need for intubation up to day 30 and the composite endpoint “need for intubation or death”. Regarding patient safety, both HFNC and NIV resulted in numerous adverse events. However, we are uncertain whether either one resulted in more adverse events due to very low certainty of the evidence. NIV resulted in considerably more serious adverse events than HFNC (7/380 vs. 0/417, *Perkins*), yet again with very low certainty of evidence. In summary, based on our analyses there is no evidence for a substantial benefit in mortality for either non-invasive respiratory therapy technique. However, NIV may result in a decreased need for endotracheal intubation, which certainly must be considered an outcome relevant to the patient.

Considering the question of benefits and harms of awake prone positioning as an additional strategy of non-invasive respiratory support for COVID-19 patients, with low certainty APP seems to have little or no effect on mortality, while there is evidence of moderate certainty that APP reduces the need for endotracheal intubation.

To make use of the presented evidence, national guideline committees could use our analyses for evidence-based decision making in the context of their local resources and requirements. Based on the patient-centered outcomes identified in this work, we consider NIV accompanied by APP to be the method of choice for respiratory support in the non-intubated COVID-19 patient with hypoxemic respiratory failure.

There is still lack of evidence with regards to overall mortality, the extent of adverse events, and for the treatment of distinct groups of patients (e.g., obesity) or pathophysiological conditions (e.g., prominent hypercapnic respiratory failure). Further high-quality trials will need to address these open questions. As it has strong implications for clinical research, we would like to stress that immanent bias in open-label studies should be countered as best as possible by increased rigor in planning, conducting, and reporting of the trials to generate useful evidence ready for meta-analysis.

## 5. Conclusions

There was no certain evidence that one of the three different non-invasive respiratory support techniques (HFNC, NIV and APP) increases or decreases mortality in patients with COVID-19 acute respiratory failure. With moderate certainty our analysis revealed that NIV compared to HFNC and APP compared to SoC decrease the need for endotracheal intubation. Given the uncertain safety profiles of NIV and HFNC, critically ill patients should be closely monitored. Irrespective of the kind of respiratory support, APP seems to be an advantageous supportive measure for non-intubated patients in respiratory failure due to COVID-19.

## Figures and Tables

**Figure 1 jcm-11-00391-f001:**
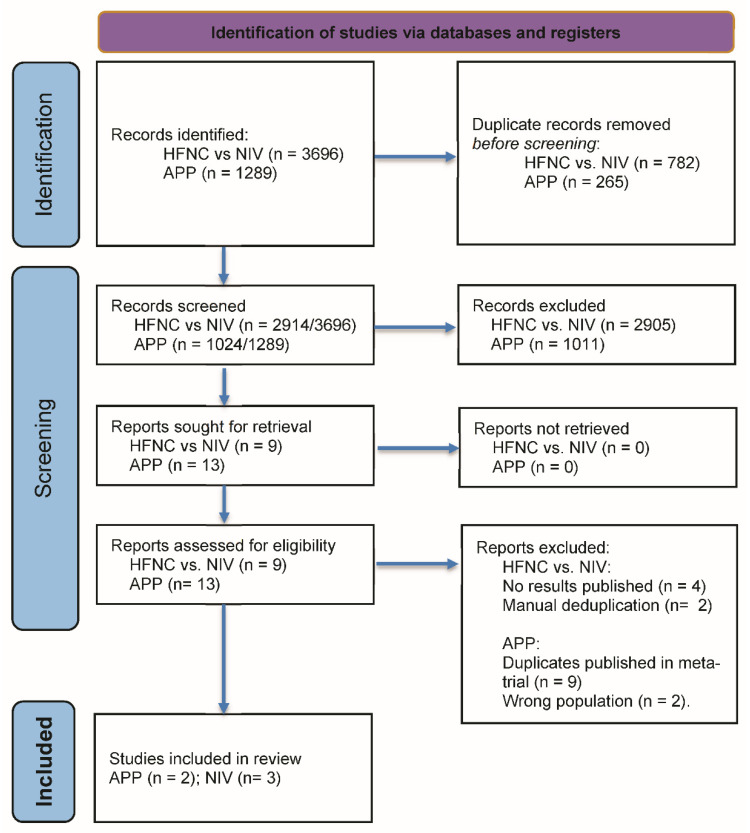
PRISMA flow diagram. HFNC: high-flow nasal cannula, NIV non-invasive ventilation, APP: awake prone-positioning; Adapted from: [24].

**Table 1 jcm-11-00391-t001:** Study characteristics. AE: adverse event, APP: awake prone-positioning, CPAP: continuous positive airway pressure, F_i_O_2_: inspiratory fraction of oxygen, HFNC: high-flow nasal cannula, ICU: intensive care unit, IQR: inter-quartile range, NIV: non-invasive ventilation, PaCO_2_: carbon dioxide partial pressure, P_a_O_2_: oxygen partial pressure, PEEP: positive end-exspiratory pressure, RR: relative risk, SD: standard deviation, WHO: world health organization.

Authors	Population	Intervention	Comparator	Outcomes	Results per Endpoint
High-flow nasal cannula vs. non-invasive ventilation					
Perkins et al. [25] Standard care in both arms: Dexamethasone and Tocilizumab added as standard care in June 2020 and January 2021, respectively Recruitment: from April 2020 to May 2021 with ca. 90% of patients recruited between September 2020 and March 2021	*n* = 797 (allocated) age (years): mean 57.6 (SD 13.0) in HFNC, 56.7 (SD 12.5) in CPAP sex: 65.2% male vs. 68.4% male comorbidities: none in 34% vs. 39%, hypertension, morbid obesity, type 2 diabetes requiring medication, chronic lung disease clinical status at enrolment: inpatients not intubated with hypoxemic respiratory failure PaO_2_/FiO_2_ ratio 139 vs. 132, resp. rate 25 vs. 26	*n* = 417 HFNC Local policies, and clinical discretion informed decisions regarding choice of device, set-up, titration, and discontinuation of treatment, mean flow 51 L/min adherence to intended treatment protocol: 92.1%	*n* = 380 NIV Local policies, and clinical discretion informed decisions regarding choice of device, set-up, titration, and discontinuation of treatment, mean PEEP 9.5 (SD 8.4) adherence to intended treatment protocol: 91.6%	Endotracheal intubation or death within 30 days In-hospital mortality 30-day mortality Endotracheal intubation within 30 days Mean length of stay in hospital (SD) Serious adverse event (as no. of patients with at least one) Adverse events (as no. of patients with at least one) Adverse events (as total no. of events per patients)	HFNC 184/414 vs. NIV 137/377: RR 1.22 (1.03 to 1.45) 88/404 vs. 72/364: RR 1.10 (0.83 to 1.45) 78/415 vs. 63/378: RR 1.13 (0.83 to 1.52) 170/414 vs. 126/377: RR 1.23 (1.02 to 1.48) 18.3 (20.0) vs. 16.4 (17.5) 0/417 vs. 7/380: RR 0.06 (0.00 to 1.06) 86/417 vs. 130/380: RR 0.60 (0.48 to 0.76) 157/417 vs. 200/380
Grieco et al. [26] Standard care in both arms: Dexamethasone 100%, Remdesivir 81%. Sedation allowed but discouraged (continuous infusion 18%HFNC group vs. 37% in the helmet group). Awake proning allowed, 60% in HFNC vs. 0% in NIV underwent proning. Other aspects according to the clinical practice in each institution. Recruitment: from October 2020 to December 2020	*n* = 109 (allocated) age (years): 63 (IQR 55–69) (intervention) 66 (IQR 57–72) (control) sex: male 84% vs. 77% comorbidities: hypertension, diabetes mell., smoking, immunocompromised state, history of cancer clinical status at enrolment: inpatients not intubated with hypoxemic respiratory failure with PaO_2_/FiO_2_ ratio equal or below 200: PaO_2_/FiO_2_ ratio 102 vs. 105, PaCO_2_ 34 vs. 34 mmHg, resp. rate 28 vs. 28	*n* = 55 HFNC for at least 48 h with 60 L/min and FiO_2_ titrated to SpO_2_ goal 92%. Stepwise weaning allowed when FiO_2_ was equal to or lower than 40% and respiratory rate lower than 25/min adherence to intended treatment protocol: received by 87% for 48 h or until intubation, one participant crossed over, other participants were weaned early according to the trial protocol	*n* = 55 Helmet NIV for at least 48 h with initial PS 10–12 mbar and PEEP of 10–12 mbar and FiO_2_ titrated to SpO_2_ goal 92%. Stepwise weaning allowed when FiO_2_ was equal to or lower than 40% and respiratory rate lower than 25/min adherence to intended treatment protocol: received by 91% for 48 h or until intubation, 4% received it at least 16 h per day, 4% interrupted treatment, 2% did not receive the allocated treatment	Days free of respiratory support within 28 days Endotracheal intubation within 28 days 28-day mortality 60-day mortality In-hospital mortality Adverse events (predefined selection of AEs relevant in respiratory and intensive care medicine, as total no. of events per patients)	Mean (SD) in HFNC vs. NIV: 15 (11) vs. 13 (11) with MD 2 days (95% CI, −2 to 6) 28/55 vs. 16/54: RR 1.72 (1.06 to 2.79) 10/55 vs. 8/54: RR 1.23 (0.52 to 2.87) 12/55 vs. 13/54: RR 0.91 (0.46 to 1.80) 14/55 vs. 13/54: RR 1.06 (0.55 to 2.04) 70/55 vs. 37/54
Nair et al. [27] Standard care in both arms: clinical management and drug therapy as per institutional protocols without further specification except: awake proning was encouraged, but only in almost all HFNC participants regularly performed. Recruitment: from August 2020 to December 2020	*n* = 109 (allocated) age (years): HFNC 57 (IQR 48–65) vs. NIV 57.5 (IQR 47–64) sex: male 80% vs. 64.8% comorbidities: Diabetes mell., hypertension, coronary heart disease, chronic kidney disease clinical status at enrolment: hospitalised, not intubated with hypoxemic respiratory failure despite O_2_ insufflation: PaO_2_/FiO_2_ ratio 105 vs. 111, PaCO_2_ 34 vs. 32 mmHg, resp. rate 30 vs. 31	*n* = 55 HFNC through large-bore binasal prongs with a high-flow heated humidifier device, initial gas flow 50 L/min and FIO_2_ of 1.0, flow and FIO_2_ were subsequently adjusted between 30–60 L/min and 0.5–1.0, to maintain SpO_2_ of 94% or more. adherence to intended treatment protocol: Not quantified, but “Increased subject compliance and ease with which awake proning could be facilitated in HFNC group could have influenced the outcome in favor of the latter.”	*n* = 54 NIV with either mask/helmet device connected to an ICU ventilator, PS of 10–20 cm H_2_O (aim of obtaining an expired tidal volume of 7–10 mL per kg of predicted body weight), PEEP 5–10 cm H2O, FIO_2_ 0.5–1.0 titrated to target SpO_2_ > 94%. adherence to intended treatment protocol: Not quantified, but “Although NIV was initiated with PS 10–20 cm H_2_O, most of the subjects required PS) 5 cm H_2_O, [...]. Subjects on NIV usually felt claustrophobic and frequently complained of dry mouth, leading to repeated detachments of the oxygenation interface. Moreover, most of the subjects on NIV were not compliant.” The incompliance with NIV could have been even higher than expected due to aggressive initial ventilator settings.	Intubation or death Intubation within 7 days In-hospital mortality Ventilator-free days at 28 d	HR 0.51 (95% CI 0.28–0.94, *p* = 0.03) 15/55 vs. 25/54: RR 0.59 (0.35 to 0.99) 16/55 vs. 25/54: RR 0.63 (0.38 to 1.04); HR 0.54 (0.29 to 1.01) median 28.0 (IQR 27–28) vs. 27.5 (27–28) These data are implausible to us, given that 27% vs. 46% of patients were intubated within 7 days and the respective Kaplan-Meier estimator in the publication. This might be due to an unreported particular definition of this complex endpoint prone to misinterpretation.
awake prone positioning					
Ehrmann et al. [28] Recruitment: 4 April 2020–26 January 2021	*n* = 1121	*n* = 564	*n* = 557	intubation or death	RR 0.86 (95% CI 0.75 to 0.98)
	age (years) 61.5 ± 13.3 (intervention) 60.7 ± 13.3 (control)	(awake) prone positioning for as long as possible	unrestricted (self) positioning except prone	28-day mortality	RR 0.87 (0.71 to 1.07)
	comorbidities diabetes mellitus, chronic heart disease, obesity			intubation within 28 days	RR 0.83 (0.71 to 0.96)
	clinical status inpatient treatment, need for HFNC or NIV, no mechanical ventilation (i.e., WHO clinical progression scale 6)			hospitalization: length of hospital stay (censored at 28 days)	difference of means −0.2 days (−1.35 to 0.96)
				skin lesions	RR 0.5 (0.16 to 1.56)
				weaning from HFNC (time to event in days)	difference of means -0.9 days (0.35 to 1.45)
Rosén et al. [29] Recruitment: 7 October 2020–7 February 2021	*n* = 75	*n* = 36	*n* = 39	intubation within 30 days	RR 1.00 (0.53 to 1.90)
	age (years) 65 (IQR 53–74; intervention) 65 (IQR 55–70; control)	(awake) prone positioning ≥16 h/day	unrestricted (self) positioning except prone	30-day mortality	RR 2.17 (0.58 to 8.03)
	comorbidities diabetes mellitus, arterial hypertension, obesity			hospitalization: length of hospital stay (censored after 30 days)	difference of means −2.0 days (−7.16 to 3.16)
	clinical status inpatient treatment, need for HFNC or NIV, no mechanical ventilation (i.e., WHO clinical progression scale 6)			skin lesions	RR 0.24 (0.06 to 1.04)
				days free of HFNC	difference of means 2.0 days (0.13 to 3.87)

**Table 2 jcm-11-00391-t002:** Risk of bias assessment of the included studies per outcome. D1: randomization process, D2: deviations from the intended interventions, D3: missing outcome data, D4: measurement of the outcome, D5: selection of the reported results, HFNC high-flow nasal cannula, NIV: non-invasive ventilation.

Study	Outcome	D1	D2	D3	D4	D5	Overall
APP
Ehrmann et al. [28]	Intubation or death at 28 days						
Ehrmann et al.	Mortality rate at 28 days						
Ehrmann et al.	Need for intubation within 28 days						
Ehrmann et al.	Hospital length of stay (censored at 28 days)						
Ehrmann et al.	Occurrence of skin lesions						
Ehrmann et al.	Weaning of HFNC (time to event)						
Rosén et al. [29]	Mortality rate at day 30						
Rosén et al.	Need for intubation within 30 days						
Rosén et al.	Occurrence of skin lesions						
Rosén et al.	Hospital length of stay (censored at 30 days)						
Rosén et al.	Days free of HFNC						
HFNC vs. NIV
Grieco et al. [26]	Mortality rate at 28 days/60 days						
Grieco et al.	In-hospital mortality						
Grieco et al.	Intubation up to day 28						
Grieco et al.	Respirator-free days at 30 days						
Grieco et al.	Hospital length of stay						
Grieco et al.	Adverse events						
Perkins et al. [25]	Intubation or death at 30 days						
Perkins et al.	In-hospital mortality						
Perkins et al.	Mortality rate at 30 days						
Perkins et al.	Hospital length of stay						
Perkins et al.	Intubation up to day 30						
Perkins et al.	Serious adverse events						
Perkins et al.	Adverse events						
Nair et al. [27]	In-hospital mortality						
Nair et al.	Intubation rate at 7 days						
Nair et al.	Intubation or death						
Nair et al.	Hospital length of stay						


 low risk, 

 some concerns, 

 high risk.

**Table 3 jcm-11-00391-t003:** Effects of HFNC vs. NIV on predefined outcomes. RR: relative risk; CI: confidence interval; HFNC: high-flow nasal cannula; NIV: non-invasive ventilation.

Outcome	Results	Absolute Effect Estimates		Certainty of Evidence
		NIV	HFNC	
Mortality: in-hospital (up to longest follow-up)	RR: 0.92 (95% CI 0.65–1.33) 986 patients from 3 studies	233 per 1000	214 per 1000	very low due to serious risk of bias, serious imprecision and indirectness
		difference: 19 less per 1000 (95% CI 82 less to 77 more)		
Mortality: up to day 30 (follow-up 28 to 30 days)	RR: 1.14 (95% CI 0.86–1.51) 902 patients from 2 studies	164 per 1000	187 per 1000	very low due to serious risk of bias, serious imprecision and indirectness
		difference: 23 more per 1000 (95% CI 23 less to 84 more)		
Intubation or death (follow-up 30 days)	RR 1.22 (95% CI 1.03–1.45) 791 patients from 1 study	363 per 1000	443 per 1000	low due to serious risk of bias and indirectness
		difference: 80 more per 1000 (95% CI 11 more to 164 more)		
Intubation (follow-up 28 to 30 days)	RR 1.34 (95% CI 1.00–1.80) 900 patients from 2 studies	329 per 1000	441 per 1000	very low due to serious risk of bias, serious imprecision and indirectness
		difference: 112 more per 1000 (95% CI 0 more to 263 more)		
Serious adverse events (follow-up 30 days)	RR 0.06 (95% CI 0.00–1.06) 797 patients from 1 study	18 per 1000	1 per 1000	very low due to serious imprecision, serious risk of bias and indirectness
		difference 17 less per 1000 (95% CI 18 less to 2 more)		
Adverse events (follow-up 30 days)	906 patients from 2 studies	Perkins: NIV 200/380 vs. HFNC 157/417 (as no. events in total) Grieco: 37/54 vs. 70/55		very low due to serious risk of bias, inconsistency and indirectness
Length of hospital stay (censored at 30 days)	difference of means +1.90 days 768 patients from 1 study	16.4 days	18.3 days	low due to serious risk of bias, serious imprecision and indirectness
		difference: 1.90 days more (95% CI 0.75 less to 4.55 more)		
Respiratory-support-free days: no invasive ventilation, HFNC, or NIV (follow-up 28 days)	difference of means 2 days 109 patients from 1 study	15 (11)	13 (11)	low due very serious imprecision
		difference: 2.0 days more (95% CI 6.16 less to 2.13 more)		

**Table 4 jcm-11-00391-t004:** Summary of findings—effects of awake prone positioning on predefined outcomes. RR: relative risk; CI: confidence interval; SoC: standard of care; APP: awake prone positioning; HFNC: high-flow nasal cannula.

Outcome	Results	Absolute Effect Estimates		Certainty of Evidence
		SoC	APP	
All-cause mortality (28 days)	RR: 1.08 (95% CI 0.51–2.31) 1196 patients from 2 studies	227 per 1000	245 per 1000	low due to serious risk of bias and serious imprecision
		difference: 18 more per 1000 (95% CI 111 less to 297 more)		
Intubation or death at 28 days	RR 0.86 (95% CI 0.75–0.98) 1121 patients from 1 study	461 per 1000	396 per 1000	moderate due to serious risk of bias
		difference: 65 less per 1000 (95% CI 115 less to 9 less)		
Intubation within 28 days	RR 0.83 (95% CI 0.71–0.96) 1196 patients from 2 studies	396 per 1000	329 per 1000	moderate due to serious risk of bias
		difference: 67 less per 1000 (95% CI 115 less to 16 less)		
Hospital length of stay (censored at 28 days)	difference of means −0.2 days 1196 patients from 2 studies	16.6 days	16.4 days	moderate due to serious risk of bias
		difference: 0.2 days less (95% CI 1.35 less to 0.96 more)		
Days free of HFNC within 30 days	difference of means 2 days 50 patients from 1 study	24 days	26 days	low due to serious risk of bias and serious imprecision
		difference: 2.0 days more (95% CI 0.13 more to 3.87 more)		
Weaning of HFNC (time to event within 28 days)	difference of means 0.9 days 1121 patients from 1 study	6.0 days	6.9 days	moderate due to serious risk of bias
		difference: 0.9 days more (95% CI 0.35 more to 1.45 more)		
Skin lesions within 28 days	RR 0.5 (95% CI 0.16–1.56) 1196 patients from 2 studies	32 per 1000	16 per 1000	very low due to serious risk of bias and very serious imprecision
		difference: 16 less per 1000 (95% CI 27 less to 18 more)		

## Data Availability

Details of the literature search are reported in the Appendix A. Extracted data and details of the risk of bias assessments are available from the authors upon request.

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
