# Peer review of "Awake Prone Positioning, High-Flow Nasal Oxygen and Non-Invasive Ventilation as Non-Invasive Respiratory Strategies in COVID-19 Acute Respiratory Failure: A Systematic Review and Meta-Analysis"

_jcm, 2022, doi:10.3390/jcm11020391_

Round 1

Reviewer 1 Report

I find this Manuscript entitled: Awake Prone Positioning, High-flow Nasal Oxygen and Non  invasive Ventilation as Non-invasive Respiratory Support in 4 COVID-19 Acute Respiratory Failure. A Systematic Review and Meta-analysis well organized. I suggest to report and describe better in the Introduction the clinical Indication of High-flow Nasal Oxygen and Non invasive Ventilation in COVID-19 patients citing this article Carter, C., Aedy, H., & Notter, J. (2020). COVID-19 disease: Non-Invasive Ventilation and high frequency nasal oxygenation. Clinics in Integrated Care1, 100006. describing also the different types of High-flow Nasal Oxygen and Non invasive Ventilation in order to introduce better these interventions.

Adjust the alignment in line 203  in the results section and also in line 22 in the effect of intervention.

Edit the references following the Journal style

Edit the name of the author in the reference n 14 following the style of the other references

Edit infection in the reference n 21 with Infection

Author Response

Dear Reviewer,

Thank you for your advice and your evaluation of our review “Awake Prone Positioning, High Flow Nasal Oxygen and Non-invasive Ventilation as Non-invasive Respiratory Support in COVID-19 Acute Respiratory Failure: A Systematic Review and Meta-analysis”.

First, we adjusted the editing according to your suggestions. Also, we described the interventions  more detailed and based on the suggested article by Carter et al. We hope it is more tangible why those interventions could be important and how they work.

We hope we were able to address your concerns as diligently and comprehensively as possible.

With the best wishes for your new year,

The authors

Reviewer 2 Report

The authors do a detailed and systemic review on the literature about non-invasive respiratory intervention for COVID. A meta-analysis was tried.

I have some comments for this manuscript.

First, I would not call APP (awake prone position) as a respiratory “support”. The title of this manuscript was improper. Second, there was heterogeneity of trials testing non-invasive respiratory intervention for COVID. This problem might not be easily be solved by a meta-analysis. Treatment for COVID is changing over time. Study design as a platform like RECOVERY is a preferred method. For example, dexamethasone or monoclonal antibodies against the spike protein of SARS-CoV-2 would modify the clinical course and these updated treatment were not added as standard of care in the beginning of COVID pandemic. Pooling trials in the different timeline would not answer the key question. Third, the ref: 26 cited by the author, was conducted in population with HFNC. The APP only might not be as good as SoC in population without HFNC.

Overall, this manuscript did not provide significant contribution to the field. I would suggest to change the format for systemic review only.

Author Response

Dear Reviewer.

Thank you for your advice and your evaluation of our review “Awake Prone Positioning, High Flow Nasal Oxygen and Non-invasive Ventilation as Non-invasive Respiratory Support in COVID-19 Acute Respiratory Failure: A Systematic Review and Meta-analysis”.

We agree that APP is less of a respiratory support than a non-invasive respiratory strategy to prevent worsening, for which reason we decided to change the title to “Awake Prone Positioning, High Flow Nasal Oxygen and Non-invasive Ventilation as Non-invasive Respiratory Strategies in COVID-19 Acute Respiratory Failure: A Systematic Review and Meta-analysis”.

Second, we understand that it is always difficult to find a good way to deal with the dynamic of the pandemic. Certain aspects of standard of care in COVID-19 treatment have been changed repeatedly over time. In our opinion, platform trials entail other problems than meta-analyses of different studies conducted in different countries (i.e. the current risk of bias assessments do not cover some potential sources of bias peculiar to platform trials, where different interventions could be tested in one individual, interventions could be compared to other interventions and not to control, adaptive allocation ratios, often encountered combination with pragmatic aspects, et cetera. An introduction to potential sources of bias in platform trials can be found here “An overview of platform trials with a checklist for clinical readers” Jay J.H. Park, https://doi.org/10.1016/j.jclinepi.2020.04.025). In the case of the included studies, recruitment timeframes were well overlapping in all included studies, co-intervetions were mostly equally distributed . We always should argue, whether evidence taken from only one platform study represents better the course of COVID-19 disease and different health systems or the comparison between studies that were conducted independently in different countries but in patients with similar disease severity. When standard of care is distributed equally between the intervention and control group (which was the case in our included studies), the impact on the treatment effect should be rather  equal as well. Moreover, study designs as open-label RCTs and timepoints of outcome assessments were sufficiently equal. Therefore, we think a meta-analysis is adequate in our review.

Third, the administration of HFNC in ref. 26 (the largest trial on APP included in this review; now ref. 27) is not an additional therapeutic intervention but merely reflects the disease severity according to the WHO clinical progression scale. Thus, the need for high-flow nasal oxygen was a prerequisite for study populations to be included in this review.

We hope we were able to address your concerns as diligently and comprehensively as possible.

With the best wishes for your new year,

The authors

Round 2

Reviewer 2 Report

The authors have improved the manuscript by addressing my concerns.